Genomic assessment reveals signal of adaptive selection in populations of the Spotted rose snapper Lutjanus guttatus from the Tropical Eastern Pacific

Mar-Silva Adán F. 1
Diaz-Jaimes Pindaro pindaro@cmarl.unam.mx 2
Domínguez-Mendoza Cristina 2
Domínguez-Domínguez Omar 3 4
Valdiviezo-Rivera Jonathan 4
Espinoza-Herrera Eduardo 5
1 Posgrado en Ciencias del Mar y Limnología, Universidad Nacional Autónoma de México, Ciudad de México , México
2 Unidad de Ecología y Biodiversidad Acuática, Instituto de Ciencias del Mar y Limnología, Universidad Nacional Autónoma de México , Mexico City , Mexico
3 Laboratorio de Biología Acuática, Facultad de Biología, Universidad Michoacana de San Nicolás de Hidalgo , Morelia , Michoacán , Mexico
4 Instituto Nacional de Biodiversidad , Quito , Ecuador
5 Dirección del Parque Nacional Galápagos , Puerto Ayora, Islas Galápagos , Ecuador
Cunha Regina
Electronic publication date: 2023 Mar 27
Publication date: 2023
Volume: 11
Electronic Location ID: e15029
Received 2022 Oct 24; Accepted 2023 Feb 17
Copyright: ©2023 Mar-Silva et al.
Copyright year: 2023
Copyright holder: Mar-Silva et al.
License: This is an open access article distributed under the terms of the Creative Commons Attribution License, which permits unrestricted use, distribution, reproduction and adaptation in any medium and for any purpose provided that it is properly attributed. For attribution, the original author(s), title, publication source (PeerJ) and either DOI or URL of the article must be cited.
License URL: https://creativecommons.org/licenses/by/4.0/

Keywords: Selection, Gene flow, Genetic divergence, Fisheries

Funding: Posgrado en Ciencias del Mar y Limnología, UNAM and by CONACYT with a PhD fellowship for CVU number 444276 The Laboratorio Nacional de Cómputo de Alto Desempeño facilitated the bioinformatic analysis using the HP system cluster platform 3000SL “Miztli” under the project LANCAD-UNAM-DGTIC-341 APC from the Instituto de Ciencias del Mar y Limnología The present work was supported by the Posgrado en Ciencias del Mar y Limnología, UNAM and by CONACYT with a PhD fellowship for CVU number 444276. The Laboratorio Nacional de Cómputo de Alto Desempeño facilitated the bioinformatic analysis using the HP system cluster platform 3000SL “Miztli” under the project LANCAD-UNAM-DGTIC-341. Support for the APC was provided by the Instituto de Ciencias del Mar y Limnología, UNAM. The funders had no role in study design, data collection and analysis, decision to publish, or preparation of the manuscript.

==============================
Background

The lack of barriers in the marine environment has promoted the idea of panmixia in marine organisms. However, oceanographic conditions and habitat characteristics have recently been linked to genetic structure in marine species. The Tropical Eastern Pacific (TEP) is characterized by dynamic current systems and heterogeneous oceanographic conditions. The Gulf of Panama (part of the equatorial segment for the TEP) is influenced by a complex current system and heterogeneous environment, which has been shown to limit the gene flow for shoreline species. Next Generation Sequencing (NGS) has contributed to detect genetic differences in previously reported panmictic species by the assessment of loci associated with selection and to understand how selection acts affects marine populations. Lutjanus guttatus is a species distributed in the TEP for which previous studies using mitochondrial data recovered a panmictic pattern along its distributional range. In this study, we used SNP data of L. guttatus individuals sampled along its range to evaluate population genetic structure and investigate whether oceanographic factors influence the species’ genetic architecture. Finally, we assessed the role of adaptive selection by evaluating the contribution of outlier and neutral loci to genetic divergence.

Methods

The RADcap method was used to obtain 24 million paired reads for 123 individuals of L. guttatus covering nearly all its distributional area. Genetic variation was assessed using both spatial and non-spatial methods by comparing three different data sets: (i) a Combined Loci (CL dataset = 2003 SNPs); a search for putative loci under selection allowed the evaluation of (ii) Neutral Loci (NL dataset = 1858 SNPs) and (iii) Outlier Loci (OL dataset = 145 SNPs). We used the estimating effective migration surface (EEMS) approach to detect possible barriers to gene flow.

Results

Genetic differences were found in the OL dataset, showing two clusters (Northern and Southern), whereas NL showed no differences. This result may be related to the Selection-Migration balance model. The limit between the Northern and Southern groups was in the Gulf of Panama, which has been previously identified as a barrier to gene flow for other species, mainly due to its heterogeneous oceanographic conditions. The results suggest that selection plays an important role in generating genetic differences in Lutjanus guttatus. A migration corridor was detected that coincides with the Costa Rica Coastal Current that flows from Central America to the Gulf of California, allowing the homogenization of the northern population. In the Southern cluster, a migration corridor was observed with the OL from Panama to Colombia, which could be associated with the currents found in the Gulf of Panama. Genetic variation found in the OL of Lutjanus guttatus highlights the usefulness of NGS data in evaluating the role of selection in population differentiation.

Introduction

The apparent absence of physical barriers to gene flow in the ocean realm has prompted the idea that most marine species are panmictic (Rocha-Olivares & Sandoval-Castillo, 2003; Munguia-Vega et al., 2018; Hernández-Álvarez et al., 2020). Large effective population sizes, a long pelagic larval phase or habitat preference have been some of the possible explanations for the lack of genetic structure in marine species (Palmerín-Serrano et al., 2021; Reguera-Rouzaud et al., 2021). However, oceanographic conditions and habitat characteristics have recently been linked to genetic structure in marine species (Weeks, 2017; Torrado et al., 2020). In particular, genetic structure has been detected for fishes distributed in the Tropical Eastern Pacific (TEP) (Sandoval-Huerta et al., 2019; Palmerín-Serrano et al., 2021; Reguera-Rouzaud et al., 2020). This biogeographic region is characterized by unstable oceanographic conditions, including temperature gradients, upwelling areas and complex current system (Robertson & Cramer, 2009; Sandoval-Huerta et al., 2019), such as the Costa Rica Coastal Current (CRCC) and the Panama Bight Gyre, which are two of the most important currents influencing dispersal of species in the TEP (Corredor-Acosta et al., 2011).

The TEP is divided in three different sections due its climatic variation, in the equatorial section of the TEP (from Costa Rica to Panama) nearshore currents and the strong oceanographic gradients make this area highly complex. Within the Equatorial section, the region known as the Gulf of Panama, which stretches west from the Panama coast to Ecuador (close to 81°W), has shown to be an important area promoting the genetic differentiation of populations. The circulation pattern in the Gulf of Panama is influenced by The Intertropical Convergence Zone (ITCZ) (Forsbergh, 1969), generates the alternation between cyclic and anticyclic gyres (Monreal Gómez, Salas de león & Aldeco Ramírez, 1999), producing a coastal current that flows toward the south (Strub & James, 2002). The patterns of circulation as well as oceanographic conditions such as temperature have been considered the possible explanation for the restricted exchange of individuals among the regions and with the north of the TEP (García-De León et al., 2018; Sandoval-Huerta et al., 2019; Reguera-Rouzaud et al., 2021). However, oceanic circulation facilitates gene flow through dispersion of larval stages between populations within the Gulf of Panama (e.g., from Panama to Colombia, as observed in Elacatinus puncticulatus (redhead goby) (Sandoval-Huerta et al., 2019)). The above mentioned patterns reveal how the role of the physical environment in genetic structuring of marine species in the TEP is critical, and constitutes an important step in understanding the origin and evolution of marine biodiversity in oceanographically complex areas.

Studies that genetically characterize populations in highly migratory species with constant gene flow have perpetuated the panmictic hypothesis in the TEP. These studies have used neutral markers, such as microsatellites (Rocha-Olivares & Sandoval-Castillo, 2003) and mitochondrial genes (Lessios & Baums, 2017; Hernández-Álvarez et al., 2020), which have limitations to identify genetic structure for these species. The use of Next Generation Sequencing (NGS) protocols have detected genetic divergence at the population level in migratory species that had previously shown panmictic populations, such as Lutjanus campechanus (Portnoy et al., 2022) and Holacanthus passer (Gatins, 2021). One of the advantages of the NGS approach is the identification of outlier loci (OL, loci putatively under selection), which provide evidence of the role of environmental factors in defining patterns of genetic divergence in a variety of organisms. The convenience of identifying environmental variables (e.g., temperature, salinity) that promote local adaptation of populations, is of relevance to understanding how adaptive selection affects the gene pool of species (Pecoraro et al., 2018), and therefore its fitness. Because of this, local adaptation is a key factor for species displaying large population sizes and considerable gene flow, allowing to reveal genetic differences due to natural selection, while other evolutionary forces (such as genetic drift) are more influenced by these biological factors (Carreras et al., 2017).

Nevertheless, the lack of enough genomic data for non-model species limits the accuracy for the identification of genes involved in adaptive selection (Milano et al., 2013). The absence of reference genomes makes it difficult to detect if more than one region is affected by selection, which is a common phenomenon. The role of many genes in selection known as polygenic selection is a complex process that allows the adaptation of the organisms and variation in their phenotypic characteristics (Rowan et al., 2021), hence the evaluation of the variation of allele frequencies of a particular region as the OL without a protein characterization, could restrict the conclusions about selection. In response to these limitations, genomes of related species can be used. This has allowed the use of outlier loci for assessment of adaptation due to selection in non-model organisms (Palumbi et al., 2019), which usually results in small-scale genetic divergence for marine species (Xu et al., 2019). Evidence of selection was reported in two sympatric species of the genus Symphodus in the Mediterranean Sea, where OL showed a clear association between genetic divergence of populations and habitat temperature (Torrado et al., 2020). Moreover, a genomic evaluation of the genetic structure of two redfish populations in the northwest Atlantic identified the existence of two ecotypes related to the deep ocean, highlighting the effect of habitat heterogeneity on marine species (Benestan et al., 2021).

Furthermore, the evaluation of the role of selection in the presence of migration, has been little assessed in marine organisms, although, migration-selection balance is a common phenomenon. The usual pattern in organisms where selection is acting at the same time as migration, is the genetic differentiation of a small proportion of the genome (usually observed with outlier loci) (Graham et al., 2018). This pattern has been recovered in populations of the blue-banded goby (Lythrypnus gilberti) distributed in the Galapagos Islands, where the oceanographic conditions have promoted an adaptive process (Bernardi, 2022). Although the growing evidence that the use of markers generated by NGS help us characterize the genetic structure of marine species, there are still very few examples of studies using NGS in the TEP (Mendoza-Portillo et al., 2020; Bernardi, 2022).

The spotted rose snapper Lutjanus guttatus is a demersal fish species displaying high dispersal capability, widely distributed in the TEP, from the Gulf of California to northern Peru, including the Galapagos Islands (Allen & Robertson, 1994; Robertson & Allen, 2015). The Spotted rose snapper is an important species for fisheries throughout the region (Rojas, Maravilla & Chicas, 2004; Sarabia-Méndez et al., 2010). Adult individuals are associated with rocky substrate, which they use as refuge areas (Del Mar Palacios & Zapata, 2014), while juveniles are found in estuaries and at the mouths of rivers. As most of lutjanid species, the Spotted rose snapper occupies shoreline habitats, and adult and juvenile movements are limited to feeding areas near to the shoreline (Hernández-Álvarez et al., 2020), oceanic migration is not reported for adults and juveniles. As a consequence, dispersal seems to be predominantly through pelagic larval dispersal and is influenced by the currents (Munguia-Vega et al., 2018). Lutjanus guttatus is listed by the International Union for Conservation of Nature (IUCN) in the “least concern” category, despite its high exploitation rate.

A recent phylogeographic study of L. guttatus and its sister species L. peru along the shoreline in the TEP using the mitochondrial D-loop region concluded that these species are genetically homogenous throughout their ranges, with high genetic diversity and signals of recent expansion (Hernández-Álvarez et al., 2020). However, this differed from a study on L. peru using microsatellite markers, which found very low but significant levels of population structure in the southern Gulf of California (Munguia-Vega et al., 2018). Another study using genetic markers with fast evolution rates detected genetic differences at the population level in L. peru and L. argentiventris throughout the TEP, where a combination of isolation by distance patterns (IBD) and oceanographic factors were responsible of the genetic structure observed (Reguera-Rouzaud et al., 2021).

In the present study, we use SNP data from specimens sampled from the continental coast distribution range of the species to evaluate the genetic variation in populations of Lutjanus guttatus in the TEP. According with previous results showing the absence of genetic differences using neutral markers across the species distribution, and considering the environmental heterogeneity and complex oceanographic characteristics of the TEP, we hypothesized that selection could play a paramount role in local adaptations and resulting in genetic differences between populations of Lutjanus guttatus across the 12,000 km sampled area along the TEP. Similarly, because of the continuous species distribution in the TEP and the pelagic larval dispersal mediated by currents, and to strengthen the results obtained in the outlier loci, we also evaluated if the IBD have an effect in the genetic differentiation of L. guttatus, as observed in other species of Lutjanus distributed in the TEP.

Material & Methods

Sample collection

A total of 192 individuals were collected from 17 locations spanning the species shoreline range (Fig. 1, Table 1). Samples were collected and preserved following procedures described in (Torres-Hernández et al., 2022). Tissue samples were stored at the Laboratorio de Genética at the Instituto de Ciencias del Mar y Limnología, Universidad Nacional Autónoma de México, and in the Icthyological Collection of the Laboratorio de Biología Acuática, of Universidad Michoacana de San Nicolas de Hidalgo, México (SEMARNAT registration number CPUM-PEC-227-07-09). Permits for specimen collection were provided by the following institutions in each country: Mexico (PPF/DGOPA-035/15; SGPA-DGVS-02920/15 and F00.DRPBCPN.DIR.RBAR-100/2015-CONANP), El Salvador (MARN-AIMA-004-2013), Panama (SC/A-17-19), Costa Rica (007-2013-SINAC and R-056-2105-OT-CONAGEBIO), and Ecuador (013/2012 PNG; N° 21-2017-EXP-CM-2016-DNB/MA and MAAE-DBI-CM-2021-0152).

Figure 1 Sampling locations and groups.

Sampling locations for Lutjanus guttatus in the Tropical Eastern Pacific. Geographical groups are color-coded. In red, the Northern group is represented by Santa Rosalía (SRO), Sonora (SON), Loreto (LOR), La Paz (LPA), Todos Santos (TSA), Sinaloa (SIN), Nayarit (NAY), Colima (COL), Michoacán (MCH), Partidito (PTO), Guerrero (GRO), Oaxaca (OAX), Salvador (SLV), and Costa Rica (CRI). In blue, the Southern group is formed by Panamá (PAN) Colombia (CLB), Ecuador (ECU).

Library preparation

The RADcap protocol (Hoffberg et al., 2016) was used to obtain genomic data. For the first step, fifteen individuals from nine sampling sites across the L. guttatus distribution range were selected to design baits for capture sequences using the 3RAD protocol to obtain genomic data (Bayona-Vásquez et al., 2019). For the 3RAD library, DNA was digested using three restriction enzymes (BamHI, ClaI and MspI) which were ligated to adapters with internal iTru indexed primer (Glenn et al., 2019). Libraries were size-selected for 500 bp ± 10% using Pippin Prep and pooled in equimolar ratios and sequenced on a HiSeq PE150 at the Oklahoma Medical Research Foundation. The reads obtained were processed in Stacks v1.42 (Catchen et al., 2011; Catchen et al., 2013), and assembled de novo to generate a FASTA file catalog of the polymorphic loci that were present in at least five of the nine populations to design the SR-Snapper-Bait. We obtained 1973 RAD-loci from which were designed baits at Arbor Biosciences. Using custom filter criteria, we selected two baits—R1 and R2—for 1000 loci which served as the pseudo-reference genome.

For the second step, we generated 3RAD libraries for 192 individuals, which were normalized to 20 ng/uL, using 5 uL per sample for a final concentration of 100 ng/uL. The samples were digested with the enzymes described above and ligated to 96 internal index adapters. Two independent pools with 96 samples per capture were constructed. For each pool a single-cycle PCR was carried out using the Kapa HiFi HotStart kit. A degenerate primer (8N) was used to index individuals for downstream filtering of PCR duplicates. After cleaning the PCR products with a 1.2:1 volume ratio of SpeedBeads:DNA, three limited-cycle PCR replicates were performed to increase the concentration (for detailed description see Hoffberg et al., 2016). This final product was run on a 1.5% agarose gel to validate the library construction. The capture process was done following the manufacturer’s protocol, using temperatures between 62.5 °C and 65 °C, selecting baits with GC content between 30% and 60%. We kept one bait for Read1 and Read2, and those that passed the filters. This set of baits was synthesized as a custom RNA myBaits® kit (for details of the RADcap protocol, see File S1).

Table 1 SNPs summary statistics.

Information and summary statistics per locality for NL and OL datasets (1858 and 145 SNPs, respectively) for Lutjanus guttatus.

Locality	Code	N	Ar	H O	H e	F IS	Ar	H O	H e	F IS	
		NL	OL	
Santa Rosalia	SRO	3	1.05	0.047	0.061	–	1.06	0.061	0.033	–	
Sonora	SON	5	1.10	0.086	0.110	0.219*	1.09	0.069	0.105	0.335*	
Loreto	LOR	3	1.09	0.059	0.104	0.432*	1.09	0.067	0.106	0.347*	
La Paz	LPA	2	1.07	0.072	0.047	–	1.11	0.095	0.175	–	
Todos Santos	TSA	5	1.07	0.056	0.084	−0.483	1.06	0.053	0.063	0.130*	
Sinaloa	SIN	15	1.11	0.081	0.071	0.326*	1.09	0.062	0.101	0.381*	
Nayarit	NAY	12	1.11	0.078	0.119	0.344*	1.09	0.058	0.093	0.371*	
Colima	COL	9	1.12	0.079	0.127	0.376*	1.10	0.066	0.113	0.405*	
Michoacan	MCH	8	1.11	0.086	0.113	0.237*	1.09	0.070	0.090	0.291*	
Partidito	PTO	3	1.09	0.075	0.102	0.265*	1.09	0.075	0.103	0.254*	
Guerrero	GRO	10	1.10	0.078	0.110	0.290*	1.10	0.067	0.110	0.386*	
Oaxaca	OAX	1	–	–	–	–	–	–	–	–	
Salvador	SLV	8	1.11	0.081	0.113	0.288*	1.10	0.059	0.105	0.192*	
Costa Rica	CRI	9	1.09	0.069	0.100	0.311*	1.10	0.078	0.105	0.253*	
Panama	PAN	12	1.10	0.078	0.110	0.291*	1.09	0.062	0.102	0.286*	
Colombia	CLB	5	1.08	0.065	0.086	0.241*	1.11	0.054	0.133	0.572*	
Ecuador	ECU	13	1.10	0.076	0.107	0.290*	1.00	0.055	0.095	0.414*	
Global		123	1.09	0.073	0.097	0.220	1.08	0.061	0.102	0.288	
Notes.

N Number of samples

Ar Allelic richness

HO Observed heterozygosity

He Expected heterozygosity

FIS Inbreeding coefficient

* Coefficient intervals above zero indicate a deficiency in heterozygosity. The genetic diversity was not calculated for Oaxaca as a single sample was examined.

Data processing

Reads of 150 bp were obtained for each pool. In Stacks v2.0 (Rochette, Rivera-Colón & Catchen, 2019) clone_filter and process_radtags modules were used to remove PCR duplicates, demultiplex samples, and trim the ligated adapters. This yielded final reads with the same length (139 bp). From this initial set of data, we retained only individuals that had at least 1000 reads and at least 4x of read depth (Hoffberg et al., 2016). Reads were mapped with the mem algorithm in BWA v 0.7.7 (Li & Durbin, 2009), using the designed baits as the reference (pseudo-genome). Finally, using the BAM files we ran the ref_ map.pl pipeline in stacks v2.0 for locus-building and SNP-calling. Finally, in Stacks v2.0, the populations module was used for retained loci presenting at least 50% of the individuals within a locality (r = 0.5), and with a maximum observed heterozygosity higher than 70% (h = 70) to exclude potential paralogues (Pedraza-Marrón et al., 2019). To filter the data we used VCFtools v0.1.15 (Danecek et al., 2011) to remove individuals with more than 20% missing data (an exception was made for localities with ≤ 5 samples, where we kept all individuals; Table S1). After the filter steps obtained a total dataset of 123 individuals which were used in the downstream analysis. From this final dataset, we removed data with allele frequencies less than 1%. We avoided local linkage disequilibrium by considering only one SNP every 1000 bases, using PLINK v. 1.9 (Purcell et al., 2007). To corroborate the absence of linkage disequilibrium, we calculated the Index of Association (ia) in R package Poppr (Kamvar, Tabima & Grünwald, 2014).

Outlier detection

We identified outlier loci using two different approaches. The first, BayeScan v.2.1 (Foll & Gaggiotti, 2008), is a Bayesian approach that identifies loci under selection using differences in allele frequencies. This method uses a linear regression to decompose FST coefficients into population-specific and locus-specific components. The BayeScan analysis was conducted using default settings, 20 pilot runs of 5,000 iterations each, burn-in of 50,000 iterations and sampling every 5,000 generations with a thinning interval of 10; individuals were grouped into localities that were considered “populations”. The second method, pcadapt was used to define loci under selection, this approach is based on Principal Component Analysis (PCA). Markers were considered to be under selection if they had a false recovery rate (FDR) correction of 0.01 in PCAdapt v.4.0.2 (Privé et al., 2020). We applied the default parameters, including α value of 0.1 and the first two PCs. All SNPs that were identified as outliers by both methods were separated, and for downstream analyses we considered three datasets: neutral loci (NL), outlier loci (OL), and all loci (combined [CL]). Sequences from the OL were blasted in the NCBI dataset (https://blast.ncbi.nlm.nih.gov/Blast.cgi) with the blastn option. Additionally, we scanned the OL dataset with the genome from L. erythropterus to evaluate if loci under selection were related with a protein (BioProjects number: PRJNA662638).

Estimates of genetic diversity

The functions basic.stats and allelic.richness from the R package hierfstat v.0.04.22 (Goudet & Jombart, 2015) were used to obtain population genetic statistics for the three datasets (CL, NL and OL). For each location we estimated the expected (He) and observed (HO) heterozygosity, inbreeding coefficient (FIS) and allelic richness (Ar). Confidence intervals for FIS values were calculated with 1000 bootstraps in the boot.ppfis function on hierfstat. Paired FST were calculated following the Weir and Cockerham method (1984) in Arlequin version 3.5.2.2 (Excoffier & Lischer, 2010). The significance level (α) was adjusted by the Benjamini & Yekutieli (2001) correction (B-Y), as proposed by Narum (2006), by dividing the critical value of α by the sum of the numbers of tests.

Assessment of population structure

We conducted a spatial analysis of molecular variance (SAMOVA) in SAMOVA 2.0 (Dupanloup, Schneider & Excoffier, 2002) considering K values from 1 to 17 (we used as the maximum K the total number of localities as recommended by Pritchard, Stephens & Donnelly, 2000), and using 10, 000 iterations without any previous grouping. The groups that best explained the genetic structure were tested by AMOVA (Analysis of molecular variance), allowing 10% missing data in ARLEQUIN version 3.5.2.2 (Excoffier & Lischer, 2010). The Discriminant Analysis of Principal Components (DAPC) was made with adegenet 2.0.1 (Jombart & Ahmed, 2011). For all datasets, this analysis was carried out in two ways—without grouping localities and grouping localities based on the SAMOVA results. The find.cluster function was used to identify the K value. The best-supported number of clusters was identified by comparison of the Bayesian Information Criterion (BIC) for the different values of K. After running the DAPC, 70 PCs, which accounted for approximately 80% of the total variation in the data set, were retained using the dapc function. Finally, the DAPC scatterplots were obtained using the ggplot2 package (Wickham, 2016) in R. To further evaluate the population structure, we used a maximum likelihood approach in ADMIXTURE 1.3.0 (Alexander & Lange, 2011). This software calculates ancestry of individuals that are unrelated using a large SNP genotype dataset. We ran ADMIXTURE under the EM algorithm as optimization method, with 2000 for bootstrap, the cross-validation procedure was used to define the most suitable K-value, and the best value of the coancestry cluster was selected based on the lowest cross-validated error (CVE).

IBD, IBE, IBR

Isolation by distance (IBD) is widely used to explain differences among populations in species with a broad geographic range (Wright, 1978). To test the IBD hypothesis we performed a Mantel test using a correlation between FST/1-FST values and the geographic distance among localities (transformed to natural logarithm (LN)). Geographic distances were calculated with marmap v1.0.4 package (Pante & Simon-Bouhet, 2013) in R. This approach considers the bathymetry of the marine environment, which can accurately determine distance while considering vertical movements of marine organisms. The values of 5 to 120 m (Robertson & Allen, 2015), were used to constrain the least cost path (LCP) algorithm. This corresponds with the depth range of habitat. Besides, mantel test was carried out using the Euclidian geographic distance, because the current could have an effect in the dispersal of pelagic larval in Lutjanid as reported in Pedraza-Marrón et al. (2019).

To test isolation by environment (IBE) and isolation by resistance (IBR), we obtained 13 oceanographic variables from Bio-ORACLE (Tyberghein et al., 2012; Assis et al., 2017). The IBE was first assessed testing the collinearity between 12 variables considering values obtained from the sampling location. A Pearson’s correlation was carried out with the R package GGally (Schloerke et al., 2018), and variables strongly correlated (>0.8) were discarded. Five oceanographic variables (temperature, salinity, primary production, phytoplankton and chlorophyll) were retained to generate the environmental matrix between localities. Environmental distances were obtained with the Canberra distance method, using the “havegdist” function from the R package vegan (Oksanen et al., 2019). For the IBR the current velocity variable was used, layer was crop to the TEP polygon size and converted to raster format using the R Package Raster (Hijmans, 2020). Resistance pairwise distance matrix was generated with Circuitscape v. 4.0.5 (McRae, 2006). The circuitscape algorithm predicts patterns of gene flow from a heterogeneous landscape, considering possible pathways between a particular region and calculating the average cumulative resistance between localities sampled (McRae & Beier, 2007). Finally, a Mantel test was performed in vegan with the Spearman’s correlation using 9999 permutations. Genetic distances were used as dependent variable and the environment and resistance were considered as independent variables.

Spatial migration

We evaluated the spatial population structure as a function of potential barriers to gene flow using EEMS (estimated effective migration surface) software (Petkova, Novembre & Stephens, 2015). This approach allowed us to differentiate between regions with strong migration rates and regions where possible barriers to gene flow exist. This method employs a matrix of genetic distances and geo-referenced samples, and a distribution polygon to determine the migration surface. For the genetic distance, a dissimilarity matrix was calculated using a bed file with the bed2diffs module from EEMS (Petkova, Novembre & Stephens, 2015). The distribution polygon was drawn following the potential habitat of L. guttatus and obtained with the polyline method in the Google Maps API v3 tool (http://www.birdtheme.org/useful/v3tool.html). For the EEMS model we used three independent runs considering 200, 300, and 450 demes, a burn-in of 1,000,000 steps and MCMC length of 5,000,000 iterations. The convergence of runs and the spatial surface (maps of effective migration rate (m)) and effective diversity (q) was visualized using the Reemsplots function in R.

Results

SNP calling and outlier detection

We obtained 24 million reads with the RADcap protocol. After filtering data and removing duplicates, 10 million paired reads were retained. The first evaluation of the quantity of reads per sample left 154 samples, which recovered 10814 SNPs with 39.64% missing data. This dataset was filtered for the missing data, generating a final dataset consisting of 123 individuals from 17 localities (Fig. 1, Table S1). From those 123 samples we retained 2003 bi-allelic SNPs in 942 loci (loci were constructed by Stacks using the baits as pseudo-genome, (as they do not have biology response, we used only to name the regions containing SNPs), which correspond to the combined loci dataset (CL; neutral and outlier). The two approaches used to detect loci putatively under selection obtained as a consensus panel of 145 candidate outlier SNPs, which were separated into the OL dataset, while another 1858 SNPs remained as neutral loci dataset (NL). Contrasting results obtained with the OL and NL datasets will be shown in the main manuscript, while results for CL are reported in File S2.

Genetic diversity

The ia indicate a lack of disequilibrium linkage in NL and OL (ṛ ¯d=0.0242, p = 0.433 and ṛ ¯d=0.0163, p = 0.84 respectively) (Fig. S1) Values of FIS for NL were almost all positive; only TSA (FIS = −0.483) had a negative value (Table 1). For the OL dataset, all FIS values were positive (ranging between 0.130 for TSA and 0.572 for CLB). Confidence intervals of FIS were above zero (Fig. S2), suggesting some degree of heterozygote deficiency (Table 1). For LPA and SRO samples, it was not possible to calculate the FIS (Table 1). The observed heterozygosity (HO) values were similar between the NL dataset (0.047–0.081) and the OL (0.053–0.078) (Table 1). For expected heterozygosity (He), NL had lower values (0.047–0.127) than OL (0.033–0.175) (Table 1). Allele richness (Ar) for NL ranged from 1.05 to 1.12, and the OL ranged from 1.06 to 1.11 (Table 1).

Population genetic structure

For the NL dataset, the results evaluating different groups were non-significant and failed to recover any geographic group (results for two groups are showed in Table S2). However, when using the outlier loci (OL) and combined loci (CL) (File S2), results based on two groups showed higher and significant FCT (0.4860; p = 0.001 and 0.0653; p = 0.001). The first group included all of the localities for the northern distribution of the species (SRO, LPA, LOR, TSA, SON, SIN, NAY, COL, MCH, PTO, GRO, OAX, SLV and CRI, hereafter known as Northern Group), and the second cluster recovered three localities from the Southern distribution (PAN, CLB and ECU, hereafter known as Southern Group) (Table S2).

Pairwise-sample FST estimates using the NL dataset were not significant for almost all the comparisons (Table 2). After adjusting the significance level for multiple testing (p = 0.015), only the comparison between SLV vs PAN was significant (FST = 0.031, p = 0.004). In contrast, the OL dataset presented higher estimations of FST, resulting in highly significant differences involving the three locations from the Southern group (Table 2): PAN (FST ranging from 0.306 to 0.498 with corresponding p-values below to 0.015), CLB (FST = 0.199–0.688, p < 0.015) and ECU (FST = 0.306–0.644, p < 0.015). None of the comparisons between these southern locations and OAX were significant, although they showed high pairwise FST; this is likely due to the fact that the OAX location consisted of a single individual.

Table 2 Pairwise-sample FST estimates.

Pairwise FST values between Lutjanus guttatus localities estimated using NL and OL datasets (1,858 and 145 SNPs, respectively).

	Northern group	Southern group	
	SRO	SON	LOR	LPA	TSA	SIN	NAY	COL	MCH	GRO	PTO	OAX	SLV	CRI	PAN	CLB	ECU	
SRO	0	−0.453	−0.650	−0.350	0.010	−0.030	−0.093	−0.903	−0.636	0.016	−0.650	0.400	−0.612	−0.198	−0.449	0.577	0.056	
SON	−0.660	0	−0.066	0.105	−0.083	0.106	0.059	0.042	0.030	0.069	−0.004	−0.575	0.040	−0.050	0.435*	0.506	0.568*	
LOR	−0.831	−0.004	0	0.234	−0.022	0.006	−0.043	−0.116	−0.066	0.018	−0.090	−0.164	−0.015	0.023	0.448*	0.688	0.644*	
LPA	0.067	−0.076	−0.044	0	0.163	0.333	0.372	0.190	0.225	0.318	0.111	0.183	0.210	0.047	0.183	0.358	0.372	
TSA	−0.464	−0.045	0.009	0.033	0	−0.059	−0.098	−0.052	−0.077	−0.093	−0.210	−0.010	−0.119	−0.085	0.320*	0.621	0.557*	
SIN	−0.658	0.005	−0.012	−0.174	−0.075	0	−0.008	0.041	0.009	−0.029	−0.012	0.155	0.009	0.035	0.498*	0.646*	0.620*	
NAY	−0.595	0.004	−0.032	−0.120	−0.056	0.008	0	0.000	−0.007	−0.025	−0.017	0.112	−0.001	0.075	0.491*	0.661*	0.634*	
COL	−0.656	−0.002	−0.014	−0.113	−0.067	−0.005	−0.001	0	−0.012	0.046	−0.093	−0.324	0.013	−0.011	0.423*	0.552*	0.561*	
MCH	−0.620	−0.009	−0.034	−0.177	−0.057	−0.011	0.001	0.000	0	0.025	−0.066	−0.061	−0.052	0.025	0.434*	0.626*	0.593*	
GRO	−0.546	0.012	−0.023	−0.155	−0.059	0.011	0.005	0.002	0.006	0	−0.009	0.042	0.018	0.010	0.487*	0.645*	0.637*	
PTO	−0.665	0.042	0.025	0.011	0.020	0.005	0.032	0.012	0.003	0.021	0	0.082	−0.089	−0.019	0.306	0.588	0.542	
OAX	−0.978	−0.136	−0.143	0.148	−0.055	−0.176	−0.188	−0.161	−0.220	−0.118	0.003	0	−0.042	−0.066	0.427	0.645	0.670	
SLV	−0.536	0.027	0.000	−0.118	−0.013	0.016	0.018	0.014	0.028	0.012	0.043	−0.117	0	0.027	0.390*	0.601*	0.557*	
CRI	−0.477	0.033	0.050	−0.007	−0.010	−0.022	−0.052	−0.019	−0.013	−0.050	0.073	0.026	−0.003	0	0.317*	0.455*	0.475*	
PAN	−0.521	0.005	−0.026	−0.121	−0.033	0.004	0.000	0.006	0.000	0.002	0.030	−0.086	0.031	−0.021	0	0.198	0.005	
CLB	−0.595	0.028	−0.006	−0.117	−0.033	−0.002	−0.002	−0.008	0.002	−0.001	0.029	−0.128	0.007	−0.035	0.001	0	0.306*	
ECU	−0.395	0.026	−0.052	−0.050	0.005	−0.035	−0.040	−0.034	−0.029	−0.038	0.063	−0.061	−0.017	−0.002	−0.015	−0.007	0	
Notes.

Values for NL below the diagonal, values for OL above the diagonal.

Significant p-values = 0.015 after Benjamini & Yekutieli adjust. P < 0.01 in bold, and p ≤ 0.001 in bold and marked with an asterisk (*).

The hierarchical AMOVAs tested the hypothesis of Northern and Southern Groups. AMOVA using the NL dataset failed to detect significant genetic differences for locations among groups (FCT = 0.005; p = 0.553). In contrast, the AMOVA using the OL dataset showed significant genetic differences for variance among Northern and Southern groups (FCT = 0.4860; p =  < 0.001) (Table 3).

Table 3 AMOVA analyses.

Results of AMOVA analyses for the NL and OL datasets (1,858 and 145 SNPs, respectively) of Lutjanus guttatus. Comparing the Northern and Southern groups recovered by SAMOVA analysis.

Two groups	Source of varation	% of variance	Fixation index	p-value	
NL	
Northern + Southern	Among groups	0.52	FCT = 0.0051	0.553	
Among populations within groups	−2.73	FSC = 0.0274	0.999	
Within populations	102.21	FST =  − 0.0221	1	
OL	
Northern + Southern	Among groups	48.61	FCT = 0.4860	<0.001	
Among populations within groups	1.37	FSC = 0.0266	0.07	
Within populations	50.02	FST = 0.4997	<0.001	

DAPC analysis recovered the Northern Group formed with the localities SRO, LPA, LOR, TSA, SON, SIN, NAY, COL, MCH, PTO, GRO, OAX, SLV and CRI, while the Southern Group was formed with localities PAN, CLB, and ECU (Fig. 2). The Admixture analysis using NL recovered K = 2 (Fig. S3A); while using OL, K = 3 was obtained (Fig. S3B) as the number of clusters that best explained the genetic subdivision. Although the better explanation for NL was 2 clusters, the plot showed no evidence of population structure (Fig. 3A). For OL the better result was the formation of three clusters, nevertheless, the results show a clear separation into of two mains groups, Northern and Southern groups remain as the most differentiated (Fig. 3B).

Figure 2 Discriminant analysis of principal components (DAPC) of Lutjanus guttatus using NL and OL datasets (1,858 and 145 SNPs, respectively).

(A) Non previous grouping versus (B) a priori grouping for NL. (C) Non previous grouping versus (D) a priori groups for OL. In red Northern Group, in blue Southern Group. Localities’ names are shown in the box.

Figure 3 Admixture graphs.

Admixture analyses using (A) NL and (B) OL datasets (1858 and 145 SNPs, respectively). Each bar represents an individual, while colors refer to the inferred membership of each K (2–3). Sampled localities and main clusters are shown at the bottom of the figure.

IBD, IBE and IBR

The Mantel test to assess the isolation by distance model failed to detect a significant correlation between genetic and geographical distances using either dataset (NL: r = 0.0587; p = 0.278 and OL: r = 0.0970; p = 0.174; Fig. S4). Second Mantel test was carried out evaluating the IBD in each of the two main groups, results do not recover any pattern of relation between the geographic distance and the genetic variation (Fig. S5).

Mantel test for the IBE and IBR for the NL, failed to detect correlation between the environment variables tested (IBE) and currents velocity (IBR) (r = 0.1241; p = 0.1158 and r =  − 0.05345; p = 0.5898 respectively) (Figs. 4A and 4C). On the other hand, mantel test for OL recovered a correlation between the genetic distance versus the oceanographic environment variables and currents velocity (r = 0.3403; p = 0.0013 and r = 0.5207; p = 0.002) (Figs. 4B and 4D).

Figure 4 Correlation of FST with environment and currents.

Mantel test to evaluate IBE and IBR in L. guttatus using (A) and (C) NL dataset (1858 SNPs) and (B) and (D) OL dataset (145 SNPs).

Spatial migration

Evaluation of the spatial genetic structure using EEMS with the NL dataset recovered patches of migration throughout the distribution area of L. guttatus in the Gulf of California (LPA and LOR) and between NAY and COL (Fig. 5A), and a migration corridor was found off the coast of CRI, which flows northward and could be associated with the Costa Rica Coastal Current (CRCC) (Fig. 5A). There was no clear barrier to migration between the Northern and Southern Groups (Fig. 5A), although migration between CRI and PAN was restricted. A barrier for gene flow was inferred off the Mexican coast (Fig. 5A).

Figure 5 Spatial structure of populations.

Model of estimated effective migration surfaces (EEMS) of Lutjanus guttatus using (A) NL and (B) OL datasets (1858 and 145 SNPs, respectively). High levels of migration rates (m) are represented in blue, while brown coloration patterns indicate that the migration rates are lower than average. Barriers to gene flow are depicted by blue arrows. PGB: Panama Bight Gyre. CRCC: Costa Rica Coastal Current.

Results for the EEMS using the OL dataset contrast with the NL, showing a possible zone that generate local adaptation between the Northern and the Southern Groups which concur with the heterogeneous environment found in the Panama Bight Gyre (Fig. 5B, Fig. S6). Like the NL results, there was a barrier between the Gulf of California localities and the Mexican Pacific shoreline (Fig. 5B). Corridors for migration were found between LOR and LPA in the Gulf of California, NAY and COL on the Mexican Pacific coast, and among PAN, CLB and ECU (Southern group). As in the NL dataset, a corridor associated with the CRCC that flows northward was inferred with the EEMS (Fig. 5B).

For the genetic diversity inferred with EEMS was similar between the NL and OL datasets. High diversity rates were observed in SRO, LOR, and LPA (Fig. S7), while there was lower diversity in CRI and GRO-MCH in the Northern group (Fig. S7). In the Southern group, PAN had the highest diversity rates, and CLB and ECU showed the lower diversity (Fig. S7).

Discussion

The present study provides the most comprehensive analysis to date of populations of Lutjanus guttatus, covering almost the entire species distribution and using genomic information. It is also the first study to detect evidence of local adaptation with loci under selection (OL). Levels of overall genetic diversity observed in the localities using the NL and OL datasets (HO = 0.073 and HO = 0.061, respectively, Table 1) were lower than the reported for other TEP marine organisms using microsatellite data (García-De León et al., 2018; Reguera-Rouzaud et al., 2021), but they are comparable with values obtained using genomic data. This is the case of the genus Jasus (J. cavereorum HO = 0.012 and J. paulensis HO = 0.087) (Silva et al., 2021), Symphodus tinca, S. ocellatus (Torrado et al., 2020), and for the freshwater species Esox lucius (Sunde et al., 2022). A proper evaluation of the lower values of heterozygosity is necessary to avoid reaching erroneous conclusions. Sunde et al. (2022) evaluated the potential that of microsatellites vs SNP data for detecting genetic structure and calculating genetic diversity; they concluded that microsatellite markers are better for characterizing genetic diversity while SNP data are better for detecting genetic structure. This conclusion could be corroborated by the differences in the heterozygosity estimates using microsatellites in L. peru (global He = 0.811) and L. argentiventris (global He = 0.757) (Reguera-Rouzaud et al., 2021), as well as our estimates using SNPs (global He = 0.097 for NL and He = 0.102 for OL). Notwithstanding the weaknesses of the SNPs for the assessment of genetic diversity, we cannot discard inbreeding to explain the deficit of heterozygotes observed in the present work as has been stated in the brook charr Salvenilus fontinalis (Castric et al., 2002). However, for a marine species with large Ne this possibility is remote, so we consider that discrete or local populations having genetic differences may result in a heterozygote deficiency due to Wahlund effect (Landínez-García et al., 2009).

Genetic variation in Lutjanus guttatus

Lutjanus guttatus is a representative species of the Lutjanidae family, known as snappers, which includes a large number of commercially important fishes (FAO, 2016). Genetic characterization of populations is essential to preserve the genetic diversity of this fishing resource. Most evaluations of population structure and phylogeography of snapper species have used mitochondrial data and have suggested panmictic populations (Garber, Tringali & Stuck, 2004; Zhang, Cai & Huang, 2006; Gomes, Sampaio & Schneider, 2012; Reguera-Rouzaud et al., 2020). Hernández-Álvarez et al. (2020) recovered a panmictic pattern for L. Peru and L. guttatus, showing the limitations of using a single genetic marker to resolve genetic structure for species with large effective population sizes and constant but probably very small gene flow, like the Snappers. The low resolution of the mitochondrial data for this group was recently evaluated in the delimitation of L. campechanus and L. purpureus, which were first considered a single species based on mitochondrial data (L. campechanus) (Gomes, Sampaio & Schneider, 2012). Nevertheless, SNP data shown to clearly segregate these two taxa into two well-differentiated species (Pedraza-Marrón et al., 2019).

Contrasting patterns of genetic differentiation that we found between neutral and outlier loci in L. guttatus are consistent with the hypothesis that the genomic data, particularly polymorphic loci as SNPs, are better for resolving genetic questions in L. guttatus where a model of selection under migration was recovered. Loci showing adaptive selection clearly separated the populations into Northern and Southern clusters (FCT = 0.4861; p < 0.001; Table 3), while no differences were found when using neutral loci. The lack of differences for neutral loci, in both mitochondrial and nuclear DNA, suggests the existence of sufficient gene flow to maintain the connectivity among populations at a large spatial scale, probably through larval dispersal as observed in pelagic species (Ward et al., 1994; Díaz-Jaimes et al., 2010). Meanwhile, the observed differences in outlier loci may be related to adaptations to environmental factors that promote genetic differences in particular loci (Graham et al., 2018). Since selection is acting in the presence of gene flow, the observed pattern should be consistent with the migration-selection equilibrium model (Graham et al., 2018; Bernardi, 2022). The migration-selection balance plays an important role in the genetic divergence of populations for marine species (Bernardi, 2022). If selection is strong enough to overcome the effects of migration, it results in differentiation at loci subjected to adaptive processes but not at neutral loci, which results in a distinct pattern determined by equilibrium between migration and genetic drift (Ribeiro, Lloyd & Bowie, 2011; Graham et al., 2018; Bernardi, 2022).

Outlier loci have been shown to be informative in detecting patterns of shallow divergence in marine species (Milano et al., 2013; Torrado et al., 2020). The evaluation of markers affected by natural selection are useful for determining the effect of oceanography on populations of marine species (Milano et al., 2013; Pecoraro et al., 2018; Torrado et al., 2020). Unfortunately, for non-model species the lack of a reference genome hinders the evaluation of the effect that selection may have on genetic structure (Milano et al., 2013). Our blast evaluation for OL failed to identify any particular protein that could be affected by the selection that we detected, although a reference genome was used. This could be due to the lack of a reference genome for the species or the method employed in the present work, as we selected a polymorphic region of the genome generating 1973 RAD loci of 280 pb long, which could have resulted in a most reduced representation of the L. guttatus genome.

The efficiency of SNPs for detecting genetic variation in marine organisms has been widely confirmed, but it is important to highlight the advantages of SNP data to analyze separately outlier and neutral loci, which allowed us to identify genetic patterns at different scales. As an example, genomic data in yellowfin tuna Thunnus albacares populations, found genetic structure among the Atlantic, Indian and Pacific oceans, but by using a set of 33 outlier loci, additional differences were detected at the intra-oceanic level for the Pacific and Indian populations (Pecoraro et al., 2018). In addition to the usefulness of SNPs for detection of genetic structure at small scale in marine organisms, outlier loci allow understand how habitat heterogeneity promotes the adaptation of organisms to environmental factors. On the other hand, demographic process determined with neutral loci allow to identify barriers for gene flow between populations (Pecoraro et al., 2018). Nevertheless, in most of the studies using neutral loci it is usually revealed weak or undetectable genetic structure (Mamoozadeh, Graves & McDowell, 2019; Maroso et al., 2021), which contrasts with the well-defined genetic differences obtained with the loci under selection. This was reported in Sparus aurata (Gilthead seabream), where neutral loci detected three genetic clusters in the Mediterranean, “but this variation was not clear and low supported”, while by using outlier SNPs there were revealed two additional populations in the eastern Mediterranean related with adaptations to environmental factors (Maroso et al., 2021). However, differences resulting from adaptive selection have limitations on elucidating the effect of demographic processes or other evolutionary forces as genetic drift and therefore on their usefulness to reconstruct the evolutionary history of populations. The contrasting results obtained with neutral and outlier loci for Lutjanus guttatus in this study reinforce the idea that demographic process as migration may be influencing our ability to detect genetic structure for the species, while there is a region of the genome influenced by adaptive selection that could be related with environmental factors. The challenge in the identification of factors contributing in the adaptation of organisms to environment lies in understanding how multigenic features are related to environment variables (Moore et al., 2014).

Oceanographic conditions influencing genetic variation

Historically, genetic structure for rocky reef fishes distributed in the TEP has been found to be strongly associated with the presence of sandy gaps in the TEP that act as barriers (Sandoval-Huerta et al., 2019; Torres-Hernández et al., 2022). However, snapper species are characterized by high gene flow mediated by pelagic larval dispersal (Hernández-Álvarez et al., 2020), and they inhabit different habitats during different life stages (juveniles individuals are found in mangroves or estuaries and adults inhabit rock substrate), making these sandy gaps less important as barriers (Reguera-Rouzaud et al., 2021). This is corroborated by our results, since the limit between the Northern and Southern genetic groups was associated with oceanographic conditions and the currents velocity (Fig. 4) more than with the presence of the sandy gaps (Fig. 5). The complex oceanographic conditions found in the Gulf of Panama have promoted endemism in the region (Robertson & Allen, 2015). The seasonal winds that generate a strong surface upwelling variation between the eastern and western Gulf of Panama have allowed variation in the growth rates of coral reef (Randall et al., 2019). Furthermore, the wind system has an effect in temperature, salinity, oxygen among other factors, promoting heterogeneous environment conditions, which have promoted genetic structure in fishes as the red-head goby (Elacatinus punticulatus) (Sandoval-Huerta et al., 2019), and in species of Lutjanus such as the red (L. guttatus) and yellow (L. argentiventris) snappers (Reguera-Rouzaud et al., 2021). Correlation found with our OL and the oceanographic condition found in the Gulf of Panama (Fig. 4B) are congruent with the hypothesis that the environment in the region is playing an important role in adaptive selection.

The barrier for the two main groups was found at the Gulf of Panama, an area where the current system and temperature are influenced by the Inter Tropical Current Zone (ITCZ). The ITCZ is located north of the equator (between 4–8°N), representing the limit between the northern and southern eastern Pacific areas, and the effect of the temperature as one of the variables that explain the genetic variation found with the OL in L. guttatus (Fig. 4) corroborated the important role for selection that the environment plays in the Gulf of Panama. Also, the ITCZ represents the southern boundary of the eastern Pacific warm pool (EPWP) located along the coast of southwestern Mexico and Guatemala, where warm, low-salinity surface water and a shallow thermocline predominate (Fiedler & Lavín, 2017), which is in accordance with the five environment variables that found to be correlated with the populations structure (Fig. 5). South of the ITCZ, the Equatorial Cold Tongue predominates, which is largely influenced by cold waters from the Peru Current and equatorial upwelling (Wyrtky, 1981). These contrasting oceanographic conditions may influence the observed differences due to adaptive selection, and gene flow through the ITCZ may be not sufficient to homogenize allele frequencies for outliers subjected to selection. The Pacific shelf of Panama is characterized by complex oceanographic patterns caused by the existence of two semi-open ocean basins with contrasting hydrography in terms of temperature, salinity and primary productivity (D’Croz & O’Dea, 2007). In this shelf, the Northern part of the Eastern Pacific Countercurrent ends, splitting into a northern and a southern branch, increasing differences in environmental factors. The southern branch flow reaches South American coasts, while the northern branch is diminished by the prominence of the Azuero Peninsula, which may limit larval dispersal toward northern areas (Fig. S7). Limited gene flow may result in the disruption of the selection-migration balance, generating the pattern of differentiation observed in outlier loci (Table 2 and Table 3).

The effect of the currents were also found with EEMS results, which recovered a barrier for the gene flow located in front of the Mexican coast (Fig. 5), Although any of the other results indicate a genetic differentiation from this locations and considering that must be taken with caution because locations from the Gulf of California have small sample size, correlation found between the genetic distance for OL and the current velocity (Figs. 4B and 4D) reinforced the idea that the current could contributed in limit movement of individuals between locations in the Northern group. In previous work, genetic structure was found in the region of the Gulf of California and Sinaloa, this genetic differentiation are related with the current system (Sandoval-Huerta et al., 2019) and habitat discontinuity (Reguera-Rouzaud et al., 2021). This suggests that future exploration with more samples and using another genomic markers are necessary to determine whether oceanographic conditions in the Northern locations are acting as a barrier for L. guttatus.

Geographic distance and migration

Geographic distance did not show an effect on genetic differences for L. guttatus populations using either NL or OL, so IBD is not contributing to any pattern of genetic structure in L. guttatus (Figs. S4 and S5). The pattern of migration recovered with results from EEMS indicates that the currents facilitate the dispersal of larvae, and that the migration is lower near to the shoreline but wider in offshore waters (Fig. 5). The high residence of adult individuals along the coast supports the dispersal through larval drift (Hernández-Álvarez et al., 2020) and broadly coincides with the migration model recovered, where corridors for migration are consistent with the trajectory of the Costa Rica Coastal Current (CRCC) (Fig. 5).

For Lutjanus guttatus, larval dispersal seems to be sufficient to homogenize populations throughout the species range for neutral loci, but not sufficient to counteract the genetic differences resulting from adaptive selection in outlier loci. Our results coincide with those observed in the blue-banded goby, where the current system from the Galapagos Archipelago allowed gene flow, recovering a panmictic pattern for the neutral loci, while results using outlier loci indicated local adaptation associated with oceanographic conditions like temperature (Bernardi, 2022).

Some studies using microsatellites in snappers have assessed the effect of oceanographic gradients and/or environmental factors on snapper species distributed in the TEP (Reguera-Rouzaud et al., 2021). Movements of adults for lutjanids are limited to the feeding areas (Hernández-Álvarez et al., 2020), while juvenile could move through the mangroves or estuaries. This general pattern of movement influenced by the current system in the TEP is not observed in the Gulf of Panama, where the currents limit dispersal for reef fishes (Sandoval-Huerta et al., 2019; Pedraza-Marrón, 2014). Currents also have showed an effect in genetic variation of lutjanids (Reguera-Rouzaud et al., 2021). Genetic studies in L. peru based on microsatellites rejected the panmixia model after analyzing the role of larval dispersal on the species’ genetic structure (Munguia-Vega et al., 2018). Likewise, recent studies with microsatellites found that oceanographic gradients and the currents have contributed to patterns of genetic structure in L. peru and L. argentiventris (Reguera-Rouzaud et al., 2021) in the TEP, this idea is according with the correlation found with the genetic variation of the OL and the current velocity of the TEP (Fig. 4D; Fig. S6). Therefore, the heterogeneous environment of the TEP could be acting as a selective pressure in snapper species.

Conservation implications

Lutjanus guttatus is an important fishery resource throughout the TEP (Hernández-Álvarez et al., 2020). In countries such as Costa Rica, Colombia and Mexico, several studies have been done to develop technologies to establish aquaculture of the species (Rojas et al., 2009; Instituto Nacional de Pesca, 2013; Abdo-De la Parra et al., 2015; Chacón-Guzmán et al., 2020). Although the genetic information used for the establishment of management units is limited, a population study carried out in the Olympia oyster concluded that the use of the SNP analysis evaluating neutral and outlier loci separately have implications in the management of fisheries species (Silliman, 2019). The overall genetic structure found in L. guttatus across the TEP seems to be related with a signal of selection, and, as reported by Silliman (2019), adaptive loci could be used for genetic monitoring protocols.

Finally, the genetic structure separating the spotted rose snapper into two main clusters (Northern and Southern groups), is important information for the conservation of this important fishery resource. The use of SNPs highlights the importance of selecting adequate markers for evaluating genetic structure in snapper, and particularly in L. guttatus, to avoid erroneous conclusions. Although the species is cataloged as least concern by the IUCN, there is an evident lack of information. The general idea of panmixia for L. guttatus remains controversial in considering that the species maintains constant gene flow. Nevertheless, the selection pattern found herein indicates that in fact the oceanographic conditions in the TEP are influencing the genetic structuring of L. guttatus. Although to date an accurate assessment for fishery status has not been made, we consider necessary to reevaluate the effect that the fishery could have on the species employing additional genomic data. With our results two clusters that are affected by selection should highlight the need of genetic data for being considered for management purposes. Further studies expanding the genome coverage would improve the genetic information for Lutjanus guttatus in order to confirm the absence of genetic structure in neutral regions and the relevance for the species of inhabiting in a heterogeneous environment, and contribute to the species conservation.

Conclusions

Two well-differentiated groups were found among populations of Lutjanus guttatus using the OL dataset, which indicates that selection is playing an important role in delineating genetic differentiation within the species. Although the search for genes associated with selection failed, correlations found between the genetic distance and the environment reinforce the hypothesis that oceanographic conditions in the Gulf of Panama have an effect on L. guttatus, as has been observed in other species. On the other hand, results for the NL dataset did not distinguish two groups, which indicates that migration allowed the homogenization of neutral loci. The contrasting results between neutral and selected loci is consistent with a Selection-Migration balance and is a step toward understanding how the oceanographic conditions affect marine populations in the presence of constant gene flow. Our analysis of effective migration surface with both datasets identified a corridor that coincided with the Costa Rica Coastal Current. For the first time, genetic structure for L. guttatus was found, highlighting the importance for choosing the appropriate marker. This leads us to propose the use of more genomic markers with the aim of evaluating whether other localities in the TEP are influenced by the oceanographic conditions (e.g., oceanographic conditions in the Gulf of California). Finally, assessment of environment variables by separated in future works for the species could help to identify which factors are most important for selection and if there are future risks for the species. This information may help to improve management programs, which is an important step toward the conservation of this important fishery resource.

Supplemental Information

Figure S1 Linkage disequilibrium

Index of Association calculated for (A) NL and (B) OL datasets (1858 and 145 SNPs, respectively).

Click here for additional data file.

Figure S2 Inbreeding CI intervals

Confidence values for the inbreeding coefficient at 95% confidence intervals for each locality.

Click here for additional data file.

Figure S3 Estimation for the number of clusters

Evaluation of the number of clusters obtained with ADMIXTURE using (A) NL and (B) OL datasets (1858 and 145 SNPs, respectively).

Click here for additional data file.

Figure S4 Isolation by Distance

Mantel test for Lutjanus guttatus using (A) NL and (B) OL datasets (1858 and 145 SNPs, respectively). Geographic distances were transformed to natural logarithm (ln).

Click here for additional data file.

Figure S5 Isolation by distance per group

Mantel test for Northern and Southern groups for Lutjanus guttatus using NL dataset (A) and (C) 1858 SNPs and OL dataset (B) and (D) 145 SNPs.

Click here for additional data file.

Figure S6 Eastern Pacific Ocean currents

Map of the oceanographic currents of the tropical Eastern pacific (TEP). Map is based on Mariano, A.J. and E.H. Ryan, 2018. http://oceancurrents.rsmas.miami.edu/. Scale bar is in °C; values shown represent long term average temperature records for April.

Click here for additional data file.

Figure S7 Spatial distribution of genetic diversity in the study area

Results for the posterior mean diversity rates (q), obtained with estimating effective migration surfaces (EEMS) analysis using (A) NL and (B) OL datasets (1858 and 145 SNPs, respectively). Higher values than average genetic diversity are presented in blue, while lower values are depicted in brown.

Click here for additional data file.

Table S1 Individuals of Lutjanus guttatus filtered using VCFT ools

Final dataset of 123 samples retained after the second filters (minor allele frequency maf = 0.01, percentage of missing data 20%).

Click here for additional data file.

Table S2 SAMOVA Results

Results of SAMOVA for the NL and OL datasets (1858 and 145 SNPs, respectively). The best representation of the observed clusters was obtained without previously defined groups.

Click here for additional data file.

File S1 SNPs detection and bait design

Identification of polymorphic loci and SR-Snapper-Bait design and synthesis.

Click here for additional data file.

File S2 Neutral and outlier combined results

Results for the 2003 SNPs corresponding to the combined loci (CL) dataset.

Click here for additional data file.

The authors would like to give special thanks to all of the people who helped with fieldwork, especially to Juan Armando Sánchez of Universidad de los Andes and Victor Piñeros for his collaboration in the collect of El Chocó, Colombia. To Arturo Angulo from University of Costa Rica and Enrique Barraza from Universidad Francisco Gavidia El Salvador. Thanks to Jairo Arroyave for helping with language editing.

Additional Information and Declarations

Competing Interests

Author Contributions

Field Study Permissions

Data Availability

The authors declare there are no competing interests.

Adán F. Mar-Silva conceived and designed the experiments, performed the experiments, analyzed the data, prepared figures and/or tables, authored or reviewed drafts of the article, and approved the final draft.

Pindaro Diaz-Jaimes conceived and designed the experiments, performed the experiments, analyzed the data, prepared figures and/or tables, authored or reviewed drafts of the article, and approved the final draft.

Cristina Domínguez-Mendoza conceived and designed the experiments, performed the experiments, analyzed the data, prepared figures and/or tables, and approved the final draft.

Omar Domínguez-Domínguez conceived and designed the experiments, prepared figures and/or tables, authored or reviewed drafts of the article, and approved the final draft.

Jonathan Valdiviezo-Rivera conceived and designed the experiments, authored or reviewed drafts of the article, sample collection, and approved the final draft.

Eduardo Espinoza-Herrera conceived and designed the experiments, authored or reviewed drafts of the article, sample collection, and approved the final draft.

The following information was supplied relating to field study approvals (i.e., approving body and any reference numbers):

Specimen collection was supported and allowed by the following institutions in México: Secretaría de Agricultura, Ganadería, Desarrollo Rural, Pesca y Alimentación (SAGARPA; PPF/DGOPA-035/15; SGPA-DGVS-02920/15 and F00.DRPBCPN.DIR.RBAR-100/2015-CONANP); Autoridad de los Recursos Acuáticos de Panamá (ARAP) in Panamá (SC/A-17-19); Ministerio de Medio Ambiente y Recursos Naturales in El Salvador (MARN-AIMA-004-2013); Ministerio de Ambiente, y Parque Nacional Galapagos in Ecuador (013/2012 PNG; N⋄21-2017-EXP-CM-2016-DNB/MA and MAAE-DBI-CM-2021-0152); Ministerio de Ambiente y Energía. Sistema Nacional de Áreas de Conservación in Costa Rica (007-2013-SINAC and R-056-2105-OT-CONAGEBIO).

The following information was supplied regarding data availability:

The raw data, filtered dataset and popmap files are available at Zenodo: Mar-Silva, Adan Fernando Mar Silva, Díaz-Jaimes, Pindaro, Domínguez-Mendoza, Cristina A., Domínguez-Domínguez, Omar, Valdiviezo-Rivera, Jonathan, & Espinoza, Eduardo. (2022). Data from Lutjanus guttatus [Data set]. In PeerJ. Zenodo. https://doi.org/10.5281/zenodo.7145441.

The sequences are available at GenBank: PRJNA892974.

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
