# Peer review of "Genomic assessment reveals signal of adaptive selection in populations of the Spotted rose snapper Lutjanus guttatus from the Tropical Eastern Pacific"

_PeerJ, doi:10.7717/peerj.15029_

## Round 0.1 · original submission · Major Revisions

The two reviewers provided detailed comments and suggestions on how to improve the article.

In my own opinion, as presented, the article has a number of weaknesses addressed by the reviewers, and below are the principal points I am concerned with:

1. Reviewer #1 indicated some concerns regarding the analyses, e.g., the lack of justification for using different K-values in DAPC and SAMOVA and the underlying hypothesis for the use of the IBD.

2. Reviewer#1 mentioned that the data deposited in zenodo cannot be accessed; that needs to be fixed.

3. According to reviewer #2, it seems that linkage was not removed from the analysis. This probably would not change the outcome but the analysis should be performed for the accuracy of calculations.

4. I agree with reviewer #2 that the term structure does not apply, as is not supported by the results based on neutral loci. The major finding of the study is the existence of loci under selection with a geographic correspondence and not the existence of genetic structure, previously undetected with a lower coverage of the genome.

Minor comment: Table 2 is not complete, as results referring to Ecuador are not visible.

·

Basic reporting

The manuscript must clearly define some concepts, for example outlier loci to moderate the scope of its conclusions.. See specific comments in my review.

Experimental design

The study lacks a well-defined study hypothesis and has some methodological aspects that will have to be reviewed and possibly redone.

Validity of the findings

I cannot review the data online with the instruction 10.5281/zenodo.7145441, please make sure that the data reported by the authors can be accessed

Additional comments

Manuscript number #78394 "Genomic assessment reveals signal of adaptive selection in populations of the spotted rose snapper Lutjanus guttatus from the Tropical Eastern Pacific", is an interesting investigation on the evaluation of genetic and genomic diversity within and between demes that uses the methodology molecular, most recent known. The objective of this study was to evaluate the population genetic structure of Lutjanus guttatus in the TEP, and to analyze how oceanographic factors influence this structure, paying special attention to the role of selection in genetic divergence. In general, the study is well executed, but my main concern for this manuscript to be accepted lies in how the authors use certain statistical methods and because of this there is a risk that the interpretation of the results will be biased. In addition to the fact that there is no clear and established hypothesis.
Below I detail some aspects that must be addressed before this study is accepted.


Introduction

Although the authors define outlier loci as presumed markers that inform about local adaptation, I think it is convenient for the general reader to mention the advantages and limitations of these loci for such purposes, especially for those life history traits linked to fitness.

Throughout the writing, I believe that it is more convenient to simply use selection than selective processes.

Lines 131-140, It is important to mention that this species is closely related to the coastline and does not show oceanic migrations, at least in juvenile and adult life stages. This may have importance on the dispersion patterns and therefore on the distribution of genetic diversity and its potential causal agents such as oceanographic currents.

Lines 153-157. Although the authors mention their objectives, the working hypothesis to be tested is not clear according to the extensive background mentioned throughout the introduction.


Material & Methods

Lines 210-227. This paragraph could be improved by simplifying the form of the SNP filter, and clearly expresses how the SNPs formed the database for subsequent statistical analyses. At the beginning of the paragraph, it is mentioned that they obtained 154 loci, but then it is commented that a total of 123. Modifying Table S1, counting the individuals, would help to understand the total database that was worked on in this study.

Lines 255-260, since this type of correction to multiple significance tests is common in the literature, I see no need to indicate the equation. On the other hand, this type of multiple test was developed for independent p values, now it has suggested useful modifications when false positives are not numerous and therefore are conservative see 10.4310/sii.2011.v4.n4.a1 (Stat Interface 2011;4(4):417–430). I advise estimating the false positive rate with these modifications. In addition, the citations of Benjamini & Yekutieli (2001) and Narum 2006 are not cited in the cited literature, check these and all the references in the manuscript.

Lines 264-272. Please justify the use of 2 to 10 K values in the SAMOVA analysis and 1 to 10 K in the DAPC analysis.

Lines 299-307. Justify the underlying hypothesis for the use of the IBD and to what extent it opposes or justifies the grouping methods such as SAMOVA, DAPC and ADMIXTURE. Consider citing specific references to the fish species studied to justify the following sentence: The values of 5 to 120 m were used to constrain the least cost path (LCP) algorithm. This corresponds with the depth range of habitat.

Note: I will not see the need to do two analyzes to find the same result, just leave SAMOVA for the definition of groups


Results
Lines 312-316. This is information repeated to what is mentioned in methods (Lines 214-224), I suggest removing this in methods and leaving this part

Lines 316-317. This sentence can lead to confusion, I suggest that the term loci be clarified. This may be different from the genetic point of view or from the bioinformatic point of view, especially if you are using the Stacks program.

Line 327. clarify if in that table are the confidence intervals or the values of Fis. In addition, if a significance test was done, and it was significant, then there is a deficit of heterozygotes

Lines 336-337, this declaration must go in methods and not in results

Lines 336-343 The test carried out in table S2 is not balanced in terms of grouping, for the OL, the southern group is made up of PAN-CLB-ECU and for the NL loci there is only CLB, so it is difficult at least with this way of annotating the grouping, decide if it is due to the types of loci or the grouping done, please check if there is any error


Lines 345-346 Change the wording, instead of saying that the values are negative, indicate that the values were not significant. For this parameter there are no negative Fst values, check the formulations to estimate Fst in the program used. Check any textbook on population genetics e.g. Principles of Population Genetics 4th Edition by Hartl & Clark

Lines 346-351 The fact that only one of the comparisons have been significant, means that they are different, here there could be a problem of false positives, since non-significant values were expected for the northern localities, so I suggest redoing these multiple comparisons test to assess the veracity of false positives.

Lines 351-357. Again, there are some comparisons other than OAX that were not significant and were expected to be. The same occurs within the comparisons with the group from the south, the comparison PAN with CLB was significant and this result was not expected. It may be necessary to find a more appropriate method for your samples to detect false positives.

Lines 365-370. These results seem repetitive, leave the DPC for the SAMOVA test in the main text and make the other analysis of individual localities a supplementary one. Write in a simple way what was found, that is, the separation of the northern and southern localities (referring to Figure 2 and the supplementary one)

Lines 372 -378. The description of genetic groups in the text (it is not clear, if there are 3 or 4), does not correspond to what is reported in Figure S1, which 5 genetic groups are expected. In addition, a lot of genetic mixture is observed between the different localities and in some of them they show genetic similarity despite the wide geographical distance, for example SON with MCH and GRO. The ADMIXTURE program must be run under several parameters, so I think it is important to consider the use of samples of unequal sizes to generate more consistent results. It is known that this type of programs is easy to use but when parameterizing incorrectly it can generate confusing results. Due to the above, I suggest running this and other programs such as FASTSTRUCTURE again, guided by reading the following two articles:
Wang, J. 2017. The computer program structure for assigning individuals to populations: Easy to use but easier to misuse. Mol. Ecol. Resource. 17: 981–990.
Liu CC, Shringarpure S, Lange K, Novembre J (2020). Exploring Population Structure with Admixture Models and Principal Component Analysis. In: Dutheil, J.Y. (eds) Statistical Population Genomics. Methods in Molecular Biology, vol 2090. Humana, New York, NY. https://doi.org/10.1007/978-1-0716-0199-0_4

Line 380. This subtitle seems strange, limiting itself to what is reported; in the following paragraph would be Isolation by distance

Line 384. In figure S2, at the log distances of 10 and 15 there is no data, what about this lag? It is possible that the structure of two groups did not allow to find a relationship between the genetic and geographical distances. Perhaps this analysis should be done for each of the two groups independently detected above, i.e., the North and the South.

Line 387 change the word migration to lowercase

Line 402 define CRCC

Line 405 There is not Figure 4C and D, please clarify…

Lines 407-408, it is difficult to observe a general pattern of increased genetic diversity from south to north, rather patches of high and low diversity are seen, especially in the northern group, please clarify what is observed in Fig S3


Discussion

Lines 411-427. The authors report certain degrees of deficit of heterozygotes for certain localities (Table 1), which needs to be interpreted and explained. I suggest reading the following article to discuss this part. This discussion should have an impact on the interpretation of the genetic structure
Heredity 89, 27–35 (2002). https://doi.org/10.1038/sj.hdy.6800089

Lines 465-472. This paragraph contains a discussion that requires reflecting on other aspects and reviewing more recent bibliography. The conclusion that the evaluation of the blast to relate the outlier loci failed to find any specific proteins may be incorrect. Due to the following
1. The method with which the authors sequenced (two types of libraries) may influence the genomic areas that were analyzed for this study, in addition, the size of the fragment used for blasting is indisputably a factor that interferes with blasting. Increasing the size of the fragment can increase the probability of finding areas of the genome with a protein.
2 It is recommended to blast with the recently published genome of a Lutjanidae (Lutjanus erythropterus) check the site https://www.ncbi.nlm.nih.gov/genome/?term=Lutjanidae

Lines 474-484 This paragraph lacks a more general discussion of the findings referred to. Indeed, there are studies where no differences are observed in the genetic structure using both types of loci, for example Hand et al. 2016; Mamoozadeh et al. 2020; Moore et al. (2014). Therefore, it might be more instructive to discuss the advantages and limitations of the method to detect outlier loci. It is particularly important to discuss the polygenic nature of many adaptive traits related to outlier loci and the claim to find a selective signal at these loci.

Hand, B. K., et al. (2016). Molecular Ecology, 25(3), 689–705. https://doi.org/10.1111/mec.13517
Mamoozadeh, N. R., et al. (2020). Evolutionary Applications, 13(4), 677–698. https://doi.org/10.1111/eva.12892
Moore, J. S., et al. (2014). Molecular Ecology, 23(23), 5680–5697. https://doi.org/10.1111/mec.12972

Lines 495-496 and 505-516. Although the authors show figures of ocean currents (Fig S4) and potential flow barriers (Fig. 4), these results do not demonstrate the relationship between both types of variables, rather it is a phenomenological description. For this, however, multivariate methods will have to be used where the correlation between explanatory variables (climatic, oceanographic, etc.) and dependent variables (genetic distances) will be evaluated. So, I suggest using some method like redundancy analysis to determine those relationships. See an example in García De León et al. 2018, cited in their study.

Lines 530-532. In this paragraph, the authors should moderate their discussion of the influence of ocean currents and the spatial pattern of genetic differentiation. I suggest putting this discussion together in a single paragraph, to avoid repeating ideas.

Lines 554-555. The work of García de León et al. (2918) is not about snappers but about Pacific hake, a species with life habits different from Lutjanus species and precisely in this, I think that the authors should pay more attention to the patterns of genetic connectivity recovered with the EEMS program. It is difficult to explain a very wide genetic flow in oceanic zones and at the same time barriers to flow in certain coastal areas, especially in a species where until now it is known that it is largely dependent on coastal ecosystems such as rocky areas, riparian tributaries, estuaries, etc. From the point of view of population genetics, a generalized dispersal of larvae in oceanic areas (zooplankton studies would help corroborate this hypothesis), as the authors claim, would avoid barriers to gene flow in coastal areas. Please clarify these aspects.

Lines 581-584. I suggest the authors moderate the wording that indicates the influence of oceanographic currents on the spatial pattern of genetic diversity, until ad hoc statistical analyzes are carried out. In addition, it would not hurt to make a description of how fishing catches are recorded in Mexico to determine how much this species is affected by this human activity, or if the fishing statistics are not specific species, which would complicate the management of fishing.


Literature
Check the spelling of scientific names in the cited literature, some are not in italics


Figure 2, I suggest modifying this figure, removing the ellipses, and leaving only the symbols with their respective north and south colors.

Figure S3 is referred after figure S4 please correct

Table 1, in Note, please change Coefficient intervals to Fis. In addition, the global values of Fis were not subjected to a significance test, why? Also, it is not clear if they are confidence intervals or the actual values of Fis, please clarify.

Check the grammar and spelling of supplemental material especially peerj-78394-File_S1, including literature

Table S1 would be more informative if the individuals that were used for the analysis were counted, in its current state it is confusing and difficult for the reader to read.
In the supplementary material file peerj-78394-File_S1 there is also a Table S1, and this is confusing, I suggest renaming tables and figures in the various supplementary files, to avoid confusion

Reviewer 2 ·

Basic reporting

The work presented here focuses on the population genetics of a TEP snapper.

Through extensive sampling across the entire distribution range and multiple countries, and based on hundreds of SNPs, the authors draw a picture using both neutral and outlier loci.

The paper is overall very interesting and worthy of publication

Experimental design

Sampling is thorough and covers the entire range of the species.

Hundreds of SNPs based on modified RAD protocols provide ample statistical power to reach cogent conclusions.

The number of SNPs is about twice the number of loci, although not explictly stated, I presume that this means that on average about two SNPs per RAD locus were used, which means that much linkage was not removed from the analysis. I doubt that results would change, but selecting single SNPs (either the first or randomly, both options are available in stacks v2) is necessary to remove redundancy and for accuracy of calculations of genetic diversity and gene flow.

Validity of the findings

Findings are very interesting, as they shows little structure with neutral markers and strong evidence of loci under selection.

The main comment about the work is that there is a confusion between structure and selection.

My understanding of the results is that neutral markers agree with previous findings that very little structure is found in this species. The novelty of this study is that, using the power of genomic data, the authors show that outlier loci are present and partition geographically.

The authors use the term structure to describe this finding, which is contrary to the neutral findings.

What is likely to happen is that, while gene flow is rampant, selection is present and local adaptation may be happening, which is seen in outlier loci.
the alternative, which is not discussed, is that panmixia is only apparent, that ancient polymorphisms are not yet sorted, and outliers show "regular' population structure. This scenario is possible but very unlikely.
In order to clarify all this, I would suggest to not use the term structure in the outlier discussion. Indeed, structure is not present, but some loci are outliers (probably due to selection/local adaptation, as is nicely explained in the discussion.

Additional comments

there are a number of typos and awkward constructions, I have written some of them here, putting in quote a suggested change:

99 Studies to characterize genetically populations in highly 'migratory' species, have perpetuated the
100 panmictic hypothesis in the TEP

102 Baums, 2017; Hern·ndez-¡lvarez et al., 2020), which have 'limitations'

106 (Portnoy et al., 2022) and Holacanthus passer (Gatins, 2021). One of 'the' advantage of the

107 NGS is the identification of outlier loci (OL, loci 'putatively' under selection)

131 The Spotted rose snapper Lutjanus guttatus is a highly 'migratory'

145 (Hern·ndez-¡lvarez et al., 2020). However, this differed from a study 'on' L. peru using

183 (Bayona-V·squez et al., 2019). For the 3RAD library, preparation 'of' DNA

316 localities (Fig. 1, Table S1). From those 123 samples we retained 2003 bi-allelic SNPs in 942
317 loci, which correspond to the combined loci dataset (CL; neutral and outlier). The two
318 approaches used to 'detect' loci putatively under selection obtained 'as' a consensus panel of 145

348 0.033 and FST = 0.42, p = 0.038 respectively), and (delete 'the') of SLV vs MCH and PAN


396 Results for the EEMS using the OL dataset contrast with the NL, showing a barrier to effective
397 Migration between the Northern and the Southern Groups related to the Panama Bight Gyre (Fig
'this is not correct, if a barrier were present you would see evidence in the NL)


It is also
413 the first study to detect genetic structure in this species. 'this is not correct, NL does not show structure, and OL simply shows evidence of local adaptation'


439 species with large effective population sizes and constant gene flow, 'the use of constant is strange, because it can be constant but very small or constant and very large'


471 identify any particular proteins that could be affected by the selective pressure that we detected,
472 but this could be due to the lack of a reference genome for the species.
'maybe but I don't think so, there are so many fish genomes, it would be very strange for a gene not to be identified at this point'

---

## Round 0.2 · accepted · Accept

This second version of the manuscript was assessed by the reviewer#1 and by myself. Considering that the authors have addressed all of the reviewers' comments, I am happy with the current version. I just found some few typos that you will have the opportunity to correct during the proof check process.

·

Basic reporting

No comment

Experimental design

No comment

Validity of the findings

No comment

Additional comments

Most of my suggestions for changes were taken care of by the authors, so I have no problem that this article should be published.